  

# The Golgi apparatus: adaptations to neuronal shape and functions

Aygul Subkhangulova [1,2,4] & Marina Mikhaylova [3,4]

## Abstract

**The Golgi apparatus is the central hub of secretory and endosomal pathways in a eukaryotic cell. Despite having a conserved basic organization, the Golgi varies greatly in structure and operation mode between different cell types, ranging from dispersed cisternae in the budding yeast to the ribbon of cisternae stacks in most mammalian cells. Cell shape and secretory demands dictate structural and functional properties of the Golgi. Neurons are a particularly interesting type of secretory cells that have a highly polarized architecture and a large and diverse secretome. The neuronal Golgi complex evolved into an elaborate set of compartmentalized organelles that process and sort diverse neuronal cargos, including synaptic proteins, neuropeptides, and neurotrophic factors. In this review, we describe the structural adaptations of the Golgi to neuronal architecture and discuss the principles of neuronal cargo sorting. We also highlight structural rearrangements of the neuronal Golgi in neurodegenerative diseases and discuss the role of mutations in Golgi-related proteins in neurodevelopment.**

**Keywords** Golgi Complex; Golgi Satellites; Trafficking; Neuronal Secretion
**Subject Categories** Cell Adhesion, Polarity & Cytoskeleton; Membranes & Trafficking; Neuroscience

## Introduction

The Golgi apparatus (also known as Golgi complex, or Golgi network) is a central organelle of the secretory pathway that ensures proper processing, sorting, and packaging of secretory cargos. This organelle was originally described in 1898 by the Italian scientist and Nobel prize laureate Camillo Golgi, best known for his seminal work on the central nervous system. In fact, the first published photograph of the Golgi complex originates from the Purkinje cerebellar neurons of the owl. This discovery was first met with skepticism, and the bumpy history of research on the Golgi complex is a topic on its own (Bentivoglio, 1998; Farquhar and Palade, 1998; Rothman, 2010).

The universal building block of the Golgi is a flattened tubular structure termed a cisterna. In most cells, several cisternae pile up in an aligned manner to form a Golgi stack. Cisternae are surrounded by the Golgi matrix containing proteins that support the organelle's structure and regulate intra-Golgi transport. In most mammalian cells, adjacent stacks are laterally fused to form the Golgi ribbon (reviewed in (Klumperman, 2011)). The morphological appearance of the Golgi is surprisingly diverse between evolutionary distinct organisms; for example, the typical ribbon structure is not present in the commonly used invertebrate models, such as the nematode and the fruit fly (Fig. 1) (Chen et al, 2006; Kondylis and Rabouille, 2009). The ribbon-like Golgi has long been considered to be restricted to vertebrates but has recently been discovered in the sea urchin and some other non-vertebrate deuterostomes, which suggests its early emergence in animal evolution and secondary loss in some lineages (Benvenuto et al, 2024).

The Golgi receives secretory cargos from the rough endoplasmic reticulum (ER), where proteins are synthesized by ER-localized ribosomes or translocated after translation in the cytosol. Thereafter, cargos move across the Golgi stack in the *cis*- to *trans*-direction. The *cis*-cisterna of the Golgi is its entrance side, which faces the ER. The ER and *cis*-Golgi are connected via a tubulovesicular structure known as the ER-Golgi intermediate compartment (ERGIC), which is morphologically similar to the Golgi but bears a distinct molecular composition (Weigel et al, 2021). During the transport across the Golgi stack, cargos are subjected to sequential processing until reaching the *trans*-Golgi network (TGN), the Golgi exit side. The TGN consists of branching tubules emanating from the *trans*-most cisterna and surrounded by many budding vesicles. At the TGN, various cargos are sorted and packaged into distinct transport carriers for further transport to final destinations (De Matteis and Luini, 2008; Stalder and Gershlick, 2020). Sorting of many cargos requires the presence of specific sorting receptors and coat proteins, which recognize amino acid motifs and/or posttranslational modifications of the cargo protein and recruit the cargo into specific transport carriers.

Several proposed models describe how secretory cargos progress across the Golgi body (Glick and Luini, 2011). Currently, the cisternal maturation model is the prevalent one. According to it, cargos travel inside the cisternae that move in the *cis*-to-*trans* direction and undergo progressive maturation (Pantazopoulou and Glick, 2019). The *cis*-cisterna closest to the ER is formed by the

[1]German Center for Neurodegenerative Diseases (DZNE), Munich, Germany. [2]School of Medicine and Health, Klinikum Rechts der Isar, Technical University of Munich, Munich, Germany. [3]AG Optobiology, Institute of Biology, Humboldt-Universität zu Berlin, Berlin, Germany. [4]These authors contributed equally: Aygul Subkhangulova, Marina Mikhaylova. ✉E-mail: aygul.subkhangulova@dzne.de; marina.mikhaylova@hu-berlin.de

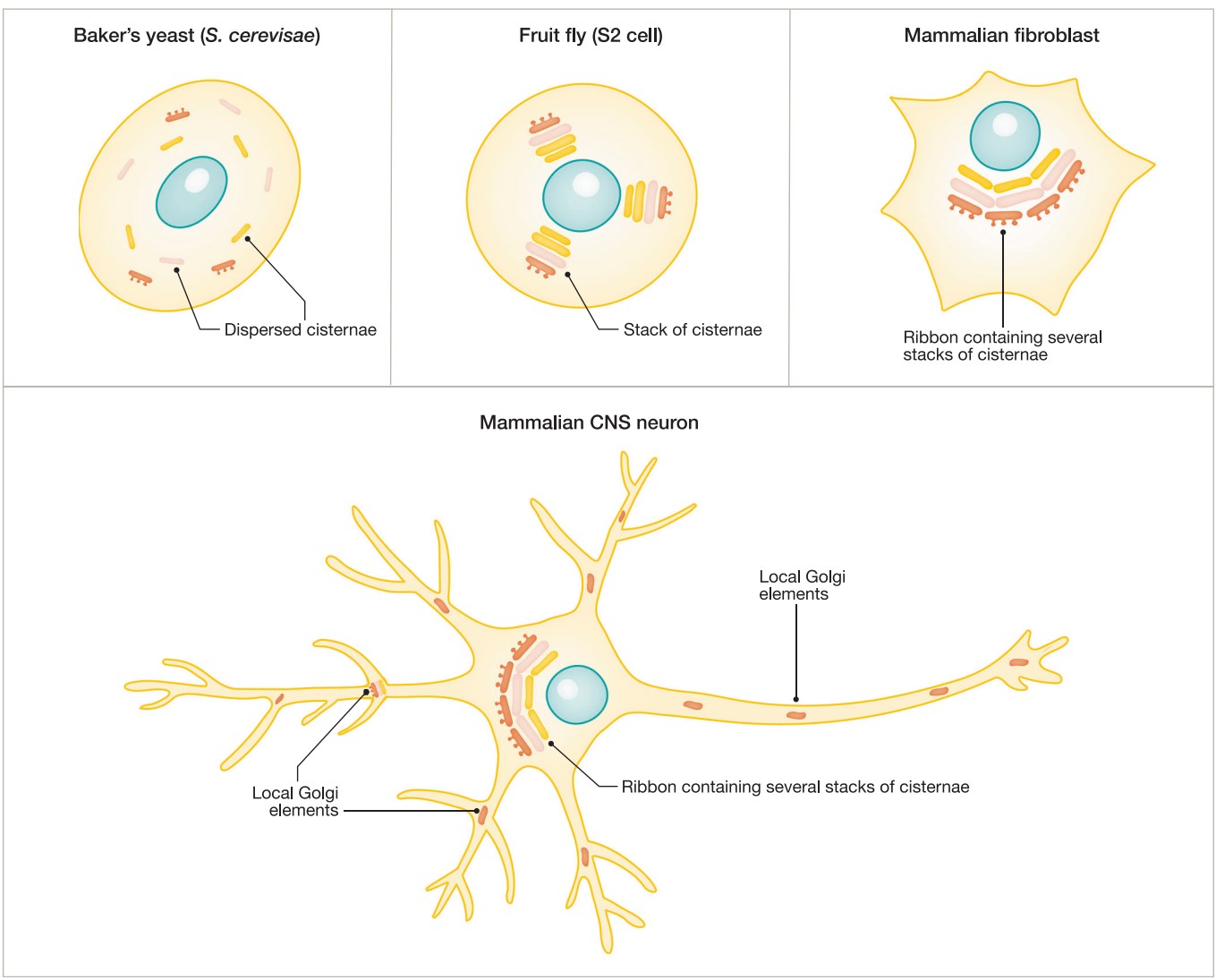

**Figure 1. Organization of the Golgi apparatus in different organisms and cell types.**

The structure of the Golgi apparatus in evolutionarily distinct organisms and cell types is shown (drawings are not to scale). Baker's yeast (*S. cerevisiae*) has Golgi in the form of isolated cisternae that are dispersed throughout the cytoplasm. Each cell harbors around 20 cisternae that differ by maturation status (*cis*-, *medial*-, *trans*-cisternae). In metazoans, Golgi cisternae are normally organized in stacks. A single stack contains the full set of cisternae needed for cargo processing. In some metazoans, e.g., fruit fly, stacks are not interconnected, while in most mammalian cells, adjacent stacks laterally fuse to form a single organelle—ribbon—that is localized around the cell nucleus. In highly polarized mammalian cells, e.g., neurons, the Golgi ribbon is often polarized and co-exists with additional Golgi elements localized to the cell periphery, such as Golgi outposts and Golgi satellites.

fusion of ER-derived vesicles, progresses through the stack, and eventually 'peels off' and disintegrates (fragments) upon budding of tubulovesicular carriers. Some cargos have been shown to move through the stack by simple diffusion via tubules that connect cisternae vertically (Beznoussenko et al, 2014).

In addition to processing and sorting of secretory cargos, the Golgi performs a number of other important functions in a eukaryotic cell, such as biosynthesis of lipids, initiation and propagation of specific signaling pathways, and nucleation of microtubules (Sanders and Kaverina, 2015; Makhoul et al, 2018; Pizzo et al, 2011). Thus, the viability and function of most eukaryotic cells rely on the functioning of the Golgi. That said, secretory cells are particularly sensitive to any Golgi

dysfunction due to the central role of this organelle in the secretory pathway.

## Unique morphology and functions of the secretory pathway in neurons

The Golgi size and position depend on the cell type and "adapt" to the current secretory demands of the cell (Clermont et al, 1995; Sengupta and Linstedt, 2011). For example, the number of cisternae per stack, as well as the size of individual cisternae, depend on the amount of secretory cargo passing through it, and can transiently increase upon an increase in cargo load (Trucco et al, 2004;

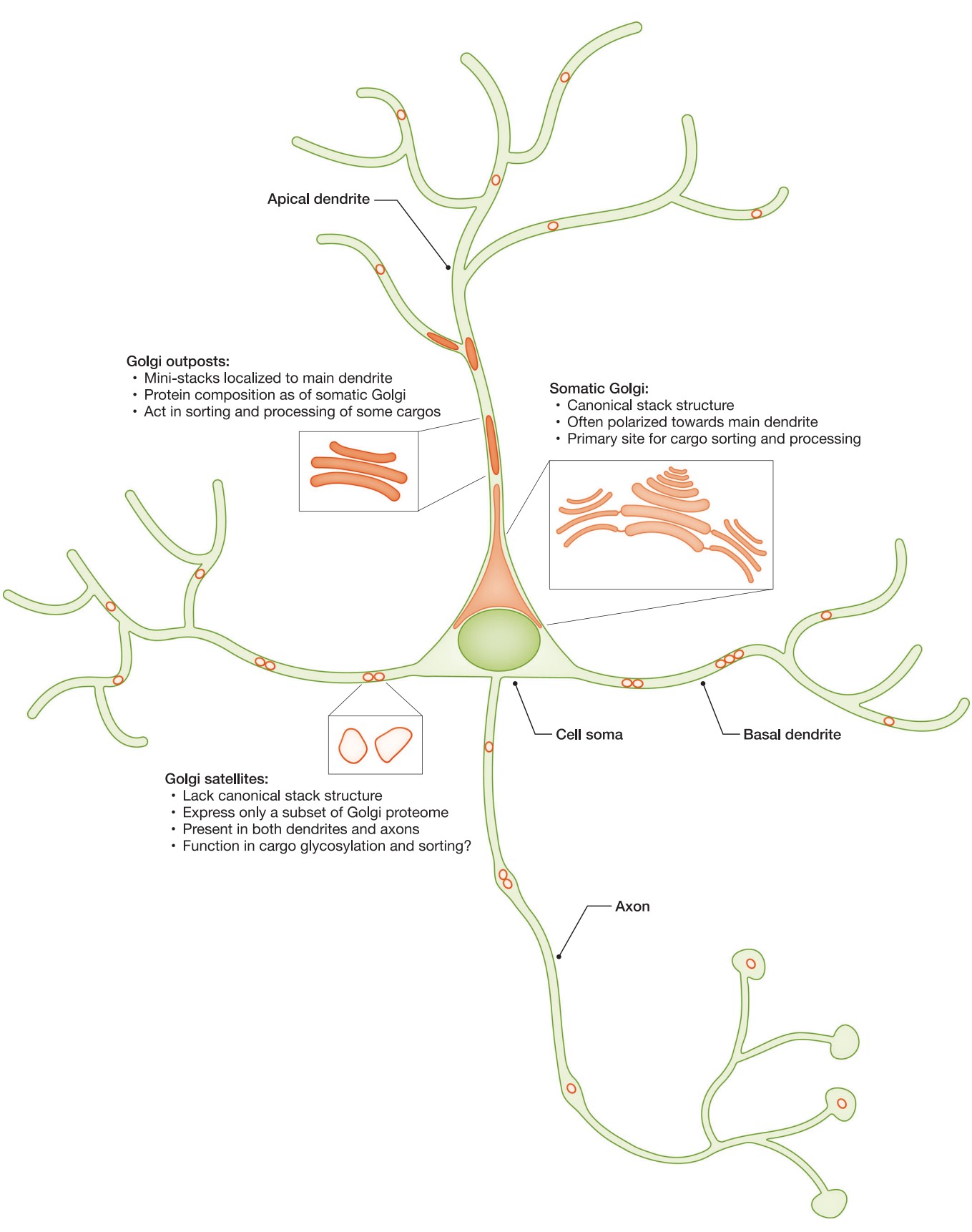

**Apical dendrite**

**Golgi outposts:**
- Mini-stacks localized to main dendrite
- Protein composition as of somatic Golgi
- Act in sorting and processing of some cargos

**Somatic Golgi:**
- Canonical stack structure
- Often polarized towards main dendrite
- Primary site for cargo sorting and processing

**Golgi satellites:**
- Lack canonical stack structure
- Express only a subset of Golgi proteome
- Present in both dendrites and axons
- Function in cargo glycosylation and sorting?

**Cell soma**

**Basal dendrite**

**Axon**

◄ **Figure 2.  Neuronal morphology and organization of the neuronal Golgi.**

Neurons are highly polarized cells that extend branched processes (neurites) far away from the cell body (soma). A typical pyramidal neuron of the rodent hippocampus has several types of neurites: a long axon, a thick apical dendrite, and multiple basal dendrites. The predominant part of the neuronal Golgi apparatus is localized to the perinuclear region of the cell soma. In addition to the somatic Golgi, neurons have smaller Golgi elements localized to neurites—Golgi outposts and Golgi satellites. This figure summarizes the main structural and functional properties of different neuronal Golgi elements (see main text for details).

Clermont et al, 1993; Rambourg et al, 1993). Neurons are a particularly interesting type of secretory cells with respect to their shape and function. A typical neuronal cell consists of a cell body (soma) and multiple neurites (axon and dendrites) that are long and extensively branched (Fig. 2). The Total dendritic length of a pyramidal neuron in the hippocampus can reach several centimeters, thereby exceeding the diameter of the cell soma by a hundredfold (Ishizuka et al, 1995). Axons of the sciatic nerves in humans can extend up to a meter. The estimated total length of the axonal arbor of dopaminergic neurons in the human substantia nigra—neurons particularly susceptible to degeneration—exceeds 4 m (Bolam and Pissadaki, 2012). Naturally, axons and dendrites require a constant supply of secretory and transmembrane cargos for growth and activity. The array of these cargos differs between axons and dendrites. Thus, neurons show an unprecedented level of cell polarization that poses a great challenge to the secretory pathway centered in the cell soma, the main site of protein synthesis. Any defect in the secretory pathway may cause irreparable damage to neurons, since these are postmitotic cells with a lifespan comparable to the lifespan of the organism.

In addition to the complex geometry and longevity, the other factors that shape the neuronal secretory pathway are the scale, versatility, and dynamic nature of the neuronal secretome. Each neuron forms thousands of synapses—interneuronal connections that act as secretory units. Neurotransmitters and neuromodulators secreted from the presynaptic side signal to the postsynapse equipped with the machinery for receiving and processing the presynaptic signals. Both pre- and postsynapses contain a large variety of secreted and transmembrane proteins, such as proteins constituting synaptic vesicles (SVs), cytomatrix of the active zone and postsynaptic density, neurotransmitter receptors, voltage-gated calcium channels, cell adhesion molecules, and many others. Recent proteomic analysis showed that purified SVs alone contain around 1500 proteins (Taoufiq et al, 2020), including transmembrane proteins required for neurotransmitter loading, vesicle acidification, docking, fusion, and subsequent endocytosis. In addition, certain molecules, such as neuropeptides and neurotrophic factors, are released extrasynaptically. Hence, neurons have an extremely high secretory output, which, in addition, is tightly regulated by the neuronal network activity.

Importantly, neurons come in a great variety of shapes and functional properties. The estimated number of neuronal types in the mammalian retina is 60, while the neocortex contains several hundred morphologically distinct neuronal types (Masland, 2004). New transcriptomics approaches have enabled further categorization of these types into thousands of subtypes with distinct molecular identities (Poulin et al, 2016; Langlieb et al, 2023). Individual neuronal types often show striking differences in size, morphology, degree of polarization, and magnitude of secretory output.

These morphological and functional features of neurons result in several remarkable adaptations of the secretory pathway in

general and the Golgi apparatus in particular (Valenzuela and Perez, 2015; Wang et al, 2020; Kennedy and Hanus, 2019; Grochowska et al, 2022). In this review, we focus specifically on the structural and functional peculiarities of the neuronal Golgi and present up-to-date evidence highlighting the critical role of the Golgi function in brain health.

## The Golgi complex in the neuronal soma

The somatic (perinuclear) Golgi is the predominant portion of the Golgi complex in neurons. It occupies a large volume of neuronal soma and has a characteristic ribbon structure conserved across many mammalian cell types. During neuronal development, the somatic Golgi undergoes a tenfold expansion, as shown in primary hippocampal neurons cultured in vitro for 12 days (Horton and Ehlers, 2003). The somatic Golgi is the primary site for processing and sorting of the neuronal secretory and membrane proteins, as well as for the biogenesis of neuron-specific secretory organelles, such as SV precursors (i.e., vesicles transporting SV proteins to presynaptic sites) and dense core vesicles (DCVs) (Zupanc, 1996; Rizalar et al, 2021).

In many neurons, including pyramidal neurons of the hippocampus, the somatic Golgi is polarized towards the apical dendrite, normally the thickest and longest dendrite of a neuron (Horton et al, 2005). During development, newly born pyramidal neurons extend multiple dynamic neurites before transitioning into a bipolar morphology during migration. Upon reaching their destination, one neurite differentiates into the axon, while another, guided by molecular cues, elongates toward the pial surface to form the apical dendrite, which later undergoes growth and branching (Barnes and Polleux, 2009). Polarized orientation of the somatic Golgi, also known as the centrosome–Golgi vector, precedes asymmetric growth of the apical dendrite in cultured hippocampal neurons and contributes to the establishment of the dendritic polarity (Horton et al, 2005; Wu et al, 2015). Similarly, the somatic Golgi is localized to the root of the primary dendrite in cerebellar Purkinje cells, and this localization precedes the dendrite specification, as shown in vivo in zebrafish brain (Tanabe et al, 2010).

Position and shape of the somatic Golgi in immature neurons are highly dynamic. In cultured mouse hippocampal neurons, the Golgi occasionally leaves the soma to invade the developing primary dendrite (Wu et al, 2015). Similar observations were made in iPSC-derived human neurons, where the somatic Golgi transiently enters one of the dendrites in the direction of neuronal migration during neuronal development (Wang et al, 2023). Moreover, it has been recently shown that in some neurons, the orientation of the Golgi may change postembryonically, and this effect is regulated by neuronal activity (Nakagawa and Iwasato, 2023). In the inhibitory spiny stellar cells of the barrel cortex, early

postnatal input leads to the lateral polarization of the somatic Golgi towards the barrel center, which in turn is required for the asymmetric dendritic projection pattern and proper responsiveness of these neurons (Nakagawa and Iwasato, 2023).

Polarization of the somatic Golgi may be important for the asymmetric growth of microtubules in axons and dendrites, as was shown recently for *Drosophila* larval dendritic arborization (*da*) neurons (Yagoubat and Conduit, 2025). In these neurons, the somatic Golgi is polarized toward the axon and serves as a microtubule-organizing center (MTOC), i.e., the structure that initiates and regulates nucleation of microtubules (Mukherjee et al, 2020). Here, microtubules grow along the *cis*-to-*trans* axis of the Golgi due to the localization of specific protein complexes to the *cis*- and *trans*-sides of the stack, and the orientation of stacks towards the axon ensures correct microtubule orientation (plus-end out) in axons. It is not clear if asymmetric microtubule nucleation depends on the Golgi orientation in mammalian neurons, but the role of the somatic Golgi as a MTOC seems to be conserved in rodent neurons (Vinopal et al, 2023), in line with its function as a MTOC in many other differentiated and undifferentiated cell types (Rios, 2014; Sanders and Kaverina, 2015).

Taken together, the orientation of the somatic Golgi plays an important role in the establishment of neuronal polarity and may instruct the direction of neuronal migration. Several mechanisms may mediate these effects. First, the position of the Golgi may affect the directionality of membrane trafficking necessary for localized cell growth. Preferential growth of the primary dendrite requires an increased delivery of the secretory cargos and membrane material, which may be favored by the orientation of the Golgi towards this dendrite. Second, polarization of the Golgi may direct organization of the neuronal microtubule cytoskeleton, which is important for the establishment of neuronal polarity. Third, polarized orientation of the somatic Golgi may affect localization of the Golgi elements—Golgi outposts—to specific dendrites, thereby favoring their outgrowth.

## Golgi elements in neurites: outposts and satellites

Considering the complexity of the neuronal shape, it is not surprising that synaptic protein synthesis occurs not only in the cell soma but also locally, in axons and dendrites (Hafner et al, 2019). Local protein synthesis may save energy and time required for the transport of the newly made proteins from the soma to synapses. After the synthesis, most proteins need to be processed and subjected to posttranslational modifications, which requires the presence of the local ER and Golgi elements. Presence and structure of the ER, both smooth (ribosome-free) and rough (ribosome-bound), along the neurites have been unequivocally documented by the 3-dimensional electron microscopy (3D-EM) (Wu et al, 2017). In contrast, much less is known about the identity and positioning of the local Golgi elements, which include Golgi outposts and Golgi satellites (Fig. 2). Proteins synthesized in neurites may be processed at the local Golgi elements, but may also travel back to the somatic Golgi for processing (Horton and Ehlers, 2003). Contribution of each of these pathways to the overall transmembrane proteome may be regulated by synaptic activity (Hanus et al, 2014). In addition, some neuronal proteins may even "bypass" the Golgi.

Golgi bypass is defined as the delivery of transmembrane proteins from the ER to the plasma membrane without those proteins passing through the Golgi apparatus (Grieve and Rabouille, 2011; Rabouille et al, 2012). This trafficking route has been first described in epithelial cells (Yoo et al, 2002; Gee et al, 2011; Hoffmeister et al, 2011) but was also proposed to exist in neurons (Bowen et al, 2017). Golgi bypass may explain the presence of proteins lacking mature glycans (i.e., core-glycosylated proteins) on the cell surface (Hanus et al, 2016), but exact mechanisms and conditions under which this trafficking route takes place remain obscure (González et al, 2018).

## Golgi outposts

Golgi outposts are isolated Golgi mini-stacks found outside of the central Golgi in some specialized cell types, such as neurons, gastric parietal cells, myocytes, and oligodendrocytes (Horton and Ehlers, 2003; Horton et al, 2005; Gunn et al, 2011; Oddoux et al, 2013; Fu et al, 2019). In neurons, Golgi outposts are localized specifically to dendrites and excluded from axons. Larger Golgi outposts (1-4 μm in diameter) are typically found in major dendrites, often in only one selected dendrite per neuron, usually the longest one. By localizing to a single dendrite, Golgi outposts may contribute to the asymmetric dendritic growth, e.g., establishment of apical dendrites in hippocampal neurons (Horton et al, 2005). Smaller Golgi outposts (0.3–1 μm in diameter) are found in minor dendrites (Quassollo et al, 2015). Around 70–80% of cultured rodent hippocampal neurons carry Golgi outposts at a given time point (Horton et al, 2005; Quassollo et al, 2015). They are often positioned in the most proximal parts of dendrites (within 30 μm away from soma) and concentrate at the dendrite branching points, but can be found as far as 100 μm away from soma (Horton et al, 2005). Most proximal Golgi outposts are often connected to the somatic Golgi and can be seen as an extension of the somatic Golgi into a dendrite. Existence of Golgi outposts in rodent neurons was validated both in primary cultures and in brain slices, by light and electron microscopy after staining of the endogenous *cis*-Golgi marker GM130 (Horton et al, 2005). In human iPSC-derived neurons, ~15–20% cells have Golgi outposts at a given time point (Wang et al, 2023). They are abundant in both proximal and distal dendritic regions of human IPSC-derived neurons and undergo fast bidirectional movements throughout the entire dendritic network (Wang et al, 2023).

Just as the Golgi stacks in the cell soma, most Golgi outpost stacks in rodent hippocampal neurons contain *cis*-, *medial*-, and *trans*-compartments and express structural Golgi proteins (e.g., GM130), as well as Golgi enzymes (e.g., sialyl- and galactosyl-transferases) (Horton and Ehlers, 2003; Horton et al, 2005; Quassollo et al, 2015). In *Drosophila* neurons, *cis*-, *medial*-, and *trans*-compartments of Golgi outposts are often disconnected and form complete stacks only at dendritic branch points (Zhou et al, 2014).

In both rodent and human neurons, Golgi outposts are formed via fission of somatic Golgi, in a process regulated by actin cytoskeleton and RhoA-ROCK signaling (Quassollo et al, 2015; Wang et al, 2023). Potential effects of neuronal activity on the Golgi outpost formation and trafficking dynamics have not been reported so far.

Functionally, Golgi outposts are thought to sort and process proteins synthesized outside of the soma, thereby acting as self-sufficient Golgi units that are independent of the somatic Golgi.

Some centrally synthesized proteins, such as NMDA receptor subunits, were also shown to traffic via Golgi outposts after exiting the soma (Jeyifous et al, 2009). Such cargos bypass the somatic Golgi and traffic within the neuronal ER system to dendrites to enter local Golgi outposts. The list of neuronal cargos that may be processed and sorted by Golgi outposts includes not only integral transmembrane proteins but also soluble cargos, such as brain-derived neurotrophic factor (BDNF) and guanine nucleotide exchange factor kalirin-7 (Horton and Ehlers, 2003; Jones et al, 2014). Trafficking of BDNF via Golgi outposts suggests that they harbor machinery needed for the formation of DCVs, since BDNF is normally packaged into DCVs for transport and release (Dieni et al, 2012).

In *da* neurons of the fly larvae, Golgi outposts were also reported to serve as microtubule nucleation sites (Ori-McKenney et al, 2012; Yalgin et al, 2015). However, loss of Golgi outposts from dendrites or genetic deletion of the known regulators of Golgi-dependent microtubule nucleation did not prevent microtubule nucleation in dendrites of these neurons (Nguyen et al, 2014; Yang and Wildonger, 2020). These manipulations resulted solely in changes in dendritic microtubule polarity, suggesting that Golgi outposts are not essential for the formation of the microtubule network but play a role in aligning microtubules via yet unknown mechanisms, as reviewed elsewhere (Valenzuela et al, 2020).

Thus, Golgi outposts represent one of the unique adaptations of the neuronal secretory system to the complex morphology of these cells. However, the relatively low abundance of neurons carrying these organelles poses a question on their functional relevance. One possibility is that current imaging methods result in the under-estimation of the true number of neurons containing Golgi outposts, since they are highly dynamic organelles that constantly traffic from and back to the soma, at least in iPSC-derived neurons. Another possibility is that Golgi outposts function at a specific time point in neuronal development, for example, during periods of neuronal growth and synaptic maturation, after which their functional importance declines. In mature neurons, other local Golgi elements (e.g., Golgi satellites), may overtake processing of the locally synthesized proteins.

## Golgi satellites

Golgi satellites are the most recently discovered and arguably least characterized components of the neuronal Golgi system. They are small (0.25–1 μm in diameter) mobile vesicular organelles found throughout the dendritic tree. Golgi satellites were originally identified in cultured rodent hippocampal neurons by overexpression of the TGN reporter probe, pGolt, which is based on the transmembrane domain of the TGN-resident protein calneuron-2 fused to the ER export sequence of the ER-to-Golgi escort protein SCAP (Mikhaylova et al, 2016). Golgi satellites are positive for some other TGN proteins, such as mannose 6-phosphate receptors, and Golgi-resident glycosylation enzymes, such as sialyltransferases (Mikhaylova et al, 2016; Andres-Alonso et al, 2023). At the same time, Golgi satellites do not contain many typical Golgi and TGN markers, such as GM130, golgin 97, and TGN38. Also, in contrast to Golgi outposts, Golgi satellites lack characteristic stack structure and appear as clusters of non-spherical vesicles, as recently shown by immuno-electron microscopy in hippocampal slices of pGolt-

transgenic mice (Andres-Alonso et al, 2023). To date, Golgi satellites were not detected in iPSC-derived human neurons (Wang et al, 2023), however the Golgi/TGN network in the referenced study was visualized by immunostainings of endogenous GM130 and p230/golgin245, which are largely absent from Golgi satellites and therefore cannot be used as reliable markers.

Originally, Golgi satellites were described in dendrites, but recent studies show their presence in axons. First, pGolt-positive structures were identified in axons of mouse peripheral neurons (Cornejo et al, 2020). More recently, axonal Golgi satellites were described in rat hippocampal neurons (Hertrich et al, 2025). In the latter study, Golgi satellites were visualized by overexpression of the fluorescently tagged sialyltransferase St3Gal5 in primary cultures. This reporter has already been used for visualization of Golgi satellites in the original study by Mikhaylova et al (2016), where it was shown to colocalize with pGolt. In the follow-up study, St3Gal5-positive structures in axons were found to be more mobile than in dendrites (Hertrich et al, 2025). Moreover, moving axonal Golgi satellites preferentially paused at presynaptic boutons, and this stalling was correlated with synaptic activity at the bouton and was mediated by the myosin VI actin motor protein (Hertrich et al, 2025). This finding suggests that neuronal activity regulates the positioning of Golgi satellites, which may reflect the increased secretory needs of active synapses.

Several late-acting Golgi glycosylation enzymes (e.g., α-mannosidase 2, N-acetylgalactosaminyltransferase 2, sialyltransferases) were localized to Golgi satellites by light microscopy, suggesting that Golgi satellites may act in protein glycosylation. To date, only two studies have tested this hypothesis, and they provide indirect evidence in support of it (Govind et al, 2021; Andres-Alonso et al, 2023). First, Golgi satellites in cultured neurons can be labeled with the wheat germ agglutinin (WGA), a protein of the lectin family that selectively binds to sialic acid and N-acetylglucosamine, complex sugars normally added in the late Golgi (Govind et al, 2021; Andres-Alonso et al, 2023). A portion of Golgi satellites takes up Helix pomatia agglutinin, a lectin that binds to O-linked glycans, also typically attached to proteins at the Golgi (Andres-Alonso et al, 2023). Second, Golgi satellites can be labeled with specific chemically modified monosaccharides that serve as building blocks in the formation of complex N- and O-sugar chains (Andres-Alonso et al, 2023). Taken together, these data indicate an enrichment of maturely glycosylated proteins in Golgi satellites, which is in line with the potential function of these organelles in cargo glycosylation (e.g., sialylation). Most interestingly, manipulation of the abundance of Golgi satellites may affect the glycosylation status of the neuronal proteome. Govind et al showed that an increase in Golgi satellite numbers caused by prolonged neuronal stimulation resulted in increased WGA uptake in cultured rodent neurons, suggesting an overall increase in maturely glycosylated proteins (Govind et al, 2021). Andres-Alonso et al showed that a decrease in Golgi satellite numbers caused by the loss of Calneuron-1 (CALN1) resulted in the impaired sialylation of the cell adhesion molecule NCAM in distal dendrites of hippocampal neurons (Andres-Alonso et al, 2023). A limitation of these studies lies in the fact that the interventions used to modulate Golgi satellite abundance (prolonged neuronal stimulation and loss of CALN1) also affect the morphology and, possibly, function of the somatic Golgi, which may contribute to the documented effects.

Newly synthesized secretory cargos transiently pass Golgi satellites after the exit from the ER, and Golgi satellites are often positioned in the proximity to the ER exit sites in dendrites (Mikhaylova et al, 2016; Govind et al, 2021; Andres-Alonso et al, 2023). Both observations are in agreement with the proposed role of Golgi satellites in the anterograde protein traffic following the exit from the ER. Furthermore, Golgi satellites may process proteins recycling from the plasma membrane, as shown by the trafficking of surface-labeled NMDA and nicotinic receptors to Golgi satellites (Mikhaylova et al, 2016; Govind et al, 2021). This finding is particularly interesting from an evolutionary perspective, since the TGN in some eukaryotes, such as the budding yeast, also serves as an early and recycling endosome (Day et al, 2018; Nagano et al, 2019). Also in plant cells, the TGN and early endosomal compartment function in close spatial association and, moreover, can exist away from the main body of the Golgi apparatus, prompting to consider the TGN as a component of the endosomal system rather than an integral part of the Golgi (Viotti et al, 2010; Uemura et al, 2014; Nakano, 2022). In mammalian macrophages, the TGN can undergo dispersal independently of the main Golgi body into vesicles positive for the early endosomal marker EEA1 (Chen and Chen, 2018). Taken together, these observations raise a question about whether Golgi satellites are organelles of a shared endosomal and TGN identity. However, typical endosomal and Golgi satellite markers rarely colocalize; instead, the majority of Golgi satellites juxtapose to early endosomes (Mikhaylova et al, 2016; Govind et al, 2021), suggesting that Golgi satellites and early endosomes are distinct organelles that are positioned adjacent to each other and likely work in close cooperation.

Mechanisms of Golgi satellite formation, as well as their full molecular composition, are unknown. As mentioned above, prolonged neuronal stimulation was shown to result in increased numbers of Golgi satellites in dendrites of cultured neurons (Govind et al, 2021). This effect coincided with the dispersal of the somatic Golgi, suggesting that Golgi satellites may derive from the fragmented perinuclear Golgi, but there is no experimental evidence supporting this hypothesis. Abundance of Golgi satellites did not increase upon neuronal stimulation in another study, although this discrepancy may result from a much shorter duration of the stimulation (2 h versus 17 h) (Skupien-Jaroszek et al, 2023). Another possible mechanism of Golgi satellite formation is their local assembly in neurites from the material delivered from the soma in the form of post-Golgi transport vesicles. Further experiments, for example, advanced live imaging approaches involving photoactivatable probes, are required to distinguish between the central (somatic) and local origin of these organelles. It is also possible that axonal and dendritic Golgi satellites differ in the mechanism of their formation and/or molecular composition.

Despite many open questions related to the identity and functions of Golgi satellites, these organelles appear to be yet another adaptation of the secretory pathway to the remarkably complex neuronal architecture. Golgi satellites may carry out selective glycosylation reactions of transmembrane cargos that are synthesized or recycled in distal neurites. Golgi satellites may also regulate local cargo sorting, as suggested by the presence of mannose 6-phosphate receptors and clathrin (Andres-Alonso et al, 2023). Numbers and positioning of Golgi satellites are dynamically regulated by neuronal activity, suggesting a role of these organelles in neuronal plasticity. Ultimately, proving the functional importance of Golgi satellites is a challenging task due to the lack of tools that specifically affect their abundance. Determination of the molecular composition of Golgi satellites may help to find proteins specific to these organelles and facilitate studies on their functional relevance.

## Sorting of neuronal cargos at the somatic Golgi

Sorting of secreted and transmembrane proteins to correct cellular destinations is one of the key functions of the Golgi. This process is accomplished via the separation of various newly synthesized proteins based on their sorting signals and the packaging of the segregated proteins into distinct transport carriers. In neurons, protein sorting is a particularly challenging task due to the large and versatile membrane proteome and secretome (Taoufiq et al, 2020), as well as the polarized cell architecture. Newly synthesized proteins need to reach many different destinations, such as presynaptic active zone, SVs, postsynaptic density, DCVs, and (endo)lysosomes.

The majority of cargos are sorted at the late Golgi and the TGN, though segregation of some cargos, for example glutamate receptor subunits, starts already at the ER (Schwenk et al, 2019). Mechanisms of neuronal cargo sorting and the type of transport carriers used for the cargo transport depend on the cargo type (Fig. 3). For example, transmembrane SV proteins, such as VAMP2, synaptophysin, SV2 and synaptotagmin-1, are sorted at the TGN into clear vesicles/vacuoles 50–300 nm in size that are transported in clusters to the presynaptic sites, as shown by immunogold electron microscopy on rodent neurons (Nakata et al, 1998; Ahmari et al, 2000; Tao-Cheng, 2007, 2020). Similarly sized vesicles and tubules transporting presynaptic proteins were observed in human iPSC-derived neurons (Rizalar et al, 2023). Light microscopy-based studies show that these carriers contain the endolysosomal marker LAMP1 and, like lysosomes, rely on the kinesin adapter Arl8 for the anterograde transport but, in contrast to lysosomes, do not contain lysosomal proteases and have a non-acidic pH (Vukoja et al, 2018; Götz et al, 2021; Rizalar et al, 2023). These data suggest that the carriers transporting SV proteins are in fact lysosome-related organelles, which raises an interesting question on how endolysosomal and presynaptic cargos are segregated at the neuronal Golgi. Late endosomes are often detected in close association with the SV cargo-carrying vesicle clusters by electron microscopy (Tao-Cheng, 2020). On the contrary, live imaging of lysosomal and SV proteins in rodent hippocampal neurons performed by another group shows that LAMP1 and SV cargos are localized to distinct carriers and rarely co-transported (De Pace et al, 2020). Thus, the nature of the SV protein transport carriers and mechanisms of their formation at the Golgi remain controversial.

Neuropeptides and neurotrophic factors are sorted at the TGN in DCVs, a type of secretory vesicle unique to neurons and neuroendocrine cells. Neuronal DCVs are around 70 nm in diameter, have a low pH and, as the name suggests, contain an electron-dense core consisting of condensed peptides (Russo, 2017; van den Pol, 2012). Several different neuropeptides can be co-packaged within the same vesicle (Salio et al, 2007). Nearly everything we know about the mechanisms of DCV formation at

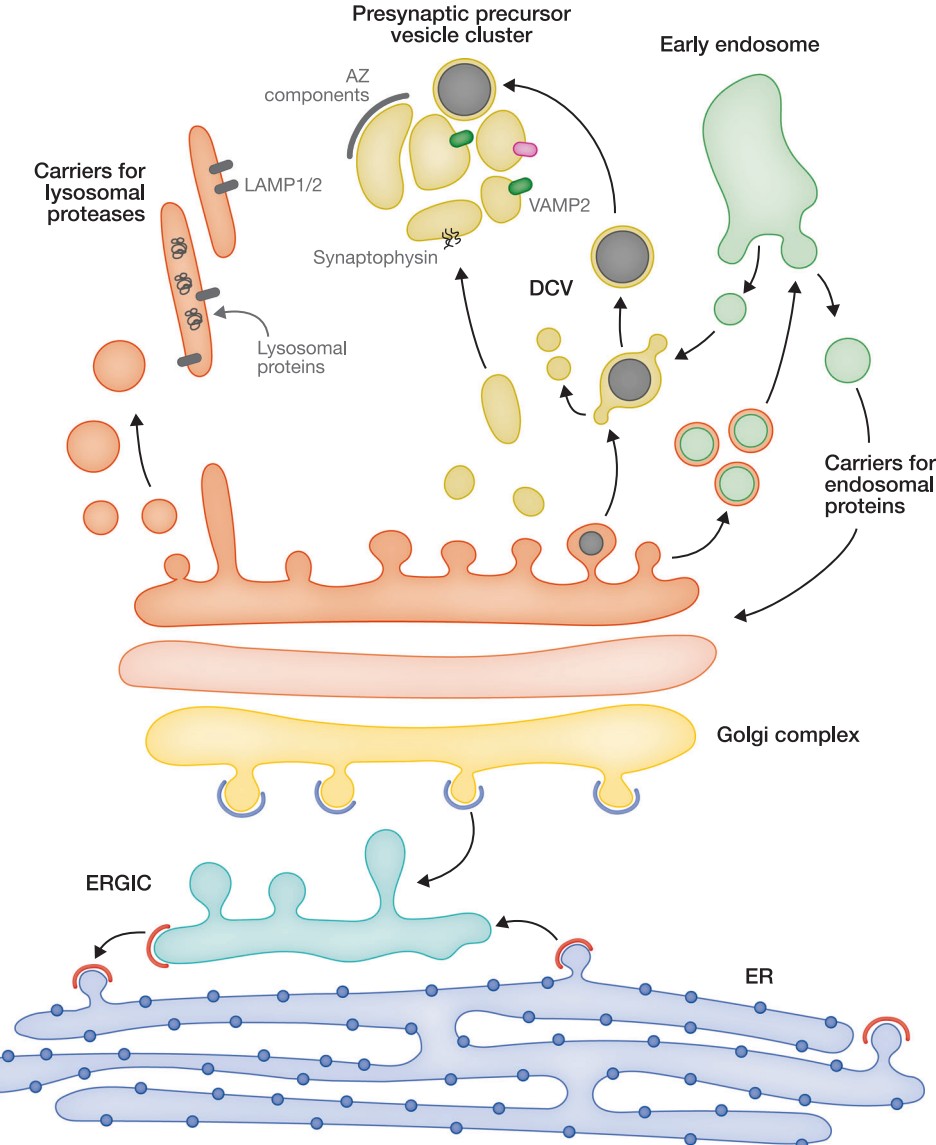

**Figure 3. Diversity of cargos sorted by the neuronal Golgi.**

The Golgi is a central hub for sorting of numerous and diverse transmembrane and secreted neuronal cargos. Most of these cargos are synthesized at the rough ER and delivered to the Golgi via the ERGIC in COPII-coated transport carriers. During the passage through the Golgi, cargos are segregated and eventually packaged at the TGN into distinct transport carriers. Transmembrane components of synaptic vesicles (SV), such as VAMP2, synaptophysin, and SV2, are sorted into clear vesicles 50–300 nm in size, which are transported from the TGN to synapses in large clusters. Neuropeptides and neurotrophic factors are sorted into dense core vesicles (DCV) with an average diameter of 70 nm. After being formed, DCV undergoes a gradual maturation process that includes condensation of the peptide cargo and removal of specific cargos required solely for vesicle maturation. DCV are often found in clusters with presynaptic precursor vesicles, possibly due to their function in transporting components of the active zone. Lysosomal proteases are sorted into LAMP1-positive transport carriers of elongated shape, which are 30–60 nm wide and up to several μm long. Other cargos sorted by the neuronal Golgi include components of the postsynaptic membrane, extrasynaptic transmembrane proteins, components of the extracellular matrix, endosomal proteins, and other proteins crucial for neuronal function.

the TGN originates from the studies on neuroendocrine and endocrine cells, such as chromaffin cells of the adrenal medulla and pancreatic beta-cells (Tooze, 1991; Morvan and Tooze, 2008; Dembla and Becherer, 2021; Castle, 2022). Neuropeptide and hormone precursors are segregated at the TGN together with granins (e.g., chromogranin A), which are acidic proteins aggregating at the low pH and relatively high calcium concentration. Condensed cargos subsequently exit the TGN in the form of immature DCVs. Immature DCVs undergo gradual maturation,

during which vesicle lumen is acidified, neuropeptide precursors are proteolytically processed to mature peptides, and certain "unwanted" cargos of the immature DCV are removed by clathrin-coated transport vesicles (Castle, 2022).

The specifics of cargo sorting to DCVs in neurons are largely unknown. In addition to the somatic Golgi, neuropeptides were shown to traffic via Golgi outposts but not via Golgi satellites, at least when overexpressed in cultured neurons (Horton and Ehlers, 2003; Mikhaylova et al, 2016). Interestingly, large scaffold proteins

of the presynaptic active zone matrix—Bassoon and Piccolo—were also shown to be transported by DCVs (Zhai et al, 2001; Shapira et al, 2003). In line with this, DCVs are present in clusters together with the presynaptic precursor vesicles (Tao-Cheng, 2020). Of note, neither Bassoon nor Piccolo passes the ER and the early Golgi due to the lack of a canonical signal peptide, but get recruited to the TGN via an unknown mechanism (Dresbach et al, 2006; Maas et al, 2012). A central core domain of Bassoon containing homo- and hetero-oligomerization sequences is sufficient to facilitate formation of DCVs at the TGN (Maas et al, 2012).

Many transmembrane synaptic proteins, for example, glutamate transporters and voltage-gated calcium channels, contain tyrosine and/or dileucine sorting motifs in their cytoplasmic tails. These sorting signals are recognized by specific coat proteins that drive the packaging of these transmembrane cargos at the TGN. Some transmembrane cargos are segregated in the early Golgi via motifs in their transmembrane and/or luminal domains (Chen et al, 2017). Soluble cargos can be sorted via interaction with transmembrane sorting receptors that serve as mediators between soluble cargos and coat proteins. For example, sorting of BDNF is mediated by sortilin, a member of the VPS10P-domain receptor family, that binds the prodomain of BDNF and ensures its packaging into DCV (Chen et al, 2005). In the absence of sortilin, BDNF is mis-sorted to constitutive secretory vesicles and to the lysosomes (Chen et al, 2005; Evans et al, 2011). Other luminal DCV cargos, for example, chromogranins, were recently suggested to be sorted into DCVs via self-assembly into condensates due to the presence of intrinsically disordered regions (Parchure et al, 2022; Campelo et al, 2023). Some of the mechanisms listed above, such as the receptor-mediated sorting, are not unique to neurons and function in most other cell types; while others, i.e., chromogranin-mediated DCV formation, are restricted to secretory cells (neurons, exocrine, endocrine, and neuroendocrine cells). Molecular machinery required for sorting is particularly diverse in neurons, and some sorting proteins are either enriched or expressed exclusively in neurons. For example, adapter protein complexes AP-1 and AP-3, components of the transport vesicle coat, though being expressed ubiquitously, have a distinct composition in neurons in comparison to other cell types (Guardia et al, 2018; Sanger et al, 2019). Two of the four subunits of the AP-3 have neuron-specific isoforms, and mutations in these subunits in mice or human patients lead specifically to neurological phenotypes in the absence of any peripheral defects (Nakatsu et al, 2004; Seong et al, 2004; Danglot and Galli, 2007; Guardia et al, 2018; Assoum et al, 2016). Thus, specialized molecular machinery for protein sorting probably reflects the complexity and importance of this process in neurons.

## Golgi and neurodevelopmental disorders

Despite the fact that the Golgi plays an essential role in almost any mammalian cell type, mutations in the Golgi-sorting machinery, Golgi-resident molecular tethers, and Golgi matrix proteins often affect the central nervous system of patients, leading to severe neurodevelopmental disorders known as "Golgipathies" (Passemard et al, 2017; El Ghouzzi and Boncompain, 2022). To date, > 40% of the Golgi-related disease-causing genes are associated with CNS disorders in humans (Zappa et al, 2018; Rasika et al, 2019). In this regard, Golgipathies are very similar to lysosomal storage

diseases caused by mutations in lysosomal proteins, which also strongly affect brain development and function. Impact of these mutations on the brain highlights a crucial role of both secretory and degradative systems for neuronal development and survival, easily explainable by the high secretory demand and postmitotic nature of neurons. In addition, defects in the Golgi function affect glial cells, most importantly oligodendrocytes, responsible for the formation of the myelin sheath.

The most common neurological symptoms of Golgipathies are postnatal-onset microcephaly (decreased brain size), myelination defects, intellectual disability, and epilepsy. In addition, patients often suffer from defects in skeletal development. An important feature of many Golgipathies is that characteristic neurological symptoms often develop early postnatally, while embryonic development is largely unaffected (as long as a given mutation is not embryonically lethal, in which case it often goes unnoticed).

One of the most characteristic golgipathies is a disorder caused by missense mutations in *ARF1* encoding a small GTPase that recruits coat proteins to the Golgi membranes and facilitates formation of transport carriers (Ge et al, 2016; Gana et al, 2022; de Sainte Agathe et al, 2023). Recently, de novo mutations in a closely related GTPase ARF3 (sharing 96% amino acid identity with ARF1) were also identified to cause brain abnormalities (Sakamoto et al, 2022; Fasano et al, 2022; El Ghouzzi and Boncompain, 2022). Overexpression of the mutant variants in non-neuronal COS-1 cells results in the Golgi fragmentation caused by a decrease in ARF stability and/or a change in the GTPase activation status (Fasano et al, 2022). Some upstream regulators of ARF GTPases, such as the two GEFs, ARFGEF1 (BIG1) and ARFGEF2 (BIG2), have also been implicated in neurodevelopmental disorders (Sheen et al, 2004; Thomas et al, 2021).

Another group of the Golgi-related proteins causing neurodevelopmental disorders are vesicle coat proteins mediating the formation of transport vesicles enriched in specific cargos. For example, mutations in the subunits of COPI complex, a coat for transport vesicles trafficking retrogradely Golgi-to-ER and within the Golgi stack, results in multi-systemic disorders characterized by developmental delay and microcephaly of variable degree in combination with a plethora of skeletal abnormalities (Izumi et al, 2016; DiStasio et al, 2017; Macken et al, 2021; Marom et al, 2021). Certain mutations in the subunits of COPII, a coat complex mediating anterograde ER-to-Golgi transport, also cause intellectual disability (Jones et al, 2003; Halperin et al, 2019). Mutations in the neuronally expressed subunits of the adapter protein complexes AP-1 and -3, or in any of the subunits of AP-4, result in severe mental retardation in combination with other neurological symptoms (Tarpey et al, 2006; Montpetit et al, 2008; Verkerk et al, 2009; Abou Jamra et al, 2011; Ammann et al, 2016; Assoum et al, 2016).

Over the years, many other Golgi-resident proteins have been implicated in neurodevelopmental disorders, as reviewed by (Rasika et al, 2019). Examples include glycosylation enzymes, tethering factors GM130 and COG (Conserved Oligomeric Golgi) complex, and lipid transfer proteins of the VPS13 family (Climer et al, 2015; Shamseldin et al, 2016; Freeze and Ng, 2011; Ugur et al, 2020). The exact molecular function of some of these proteins remains a topic of intense investigations, which are expected to shed light on the cellular causes of the associated diseases. Importantly, the function of many of the aforementioned proteins

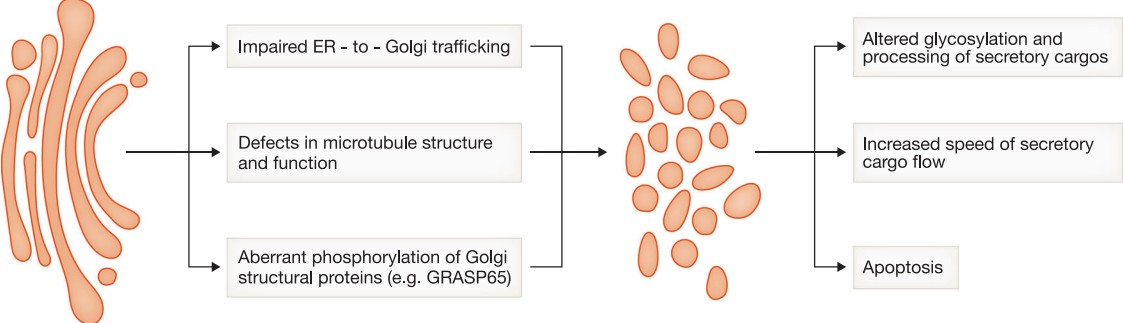

**Figure 4. Possible mechanisms and consequences of Golgi fragmentation in neurodegeneration.**

Fragmentation of the Golgi in neurons is a typical cellular feature of many neurodegenerative diseases. Mechanisms leading to the Golgi fragmentation are disease-specific but fall into several categories depicted on the scheme, along with the possible effects of the Golgi fragmentation on neuronal function and health. Golgi icons provided by Servier Medical Art (https://smart.servier.com/), licensed under CC-BY 3.0.

is not restricted to the Golgi (e.g., ARF proteins mediate coat recruitment not only to the Golgi but also to endosomal membranes), and mutations in these proteins are likely to cause multiple defects in both secretory and endosomal pathways.

## Fragmentation of the neuronal Golgi in health and disease

An important feature of the Golgi apparatus is its structural plasticity. Under specific conditions, the Golgi can lose its characteristic ribbon structure and be disassembled into mini-stacks and clusters of vesicles. This process, known as Golgi fragmentation or dispersion, is observed during cell division in mitotic cells (Wei and Seemann, 2017). Neurons, as postmitotic cells, do not undergo mitosis-induced Golgi disassembly but, strikingly, show dispersion of the Golgi stacks during periods of high neuronal activity. For example, primary hippocampal neurons undergo Golgi fragmentation upon treatment with increased potassium concentrations, inhibition of GABA receptors, or activation of NMDA receptors, in a calcium-dependent manner (Nakagomi et al, 2008; Thayer et al, 2013; Kaneko et al, 2016; Mohamed et al, 2017). Golgi fragmentation was also observed in patients with epilepsy, as well as in the hippocampi of rats after kainic acid-induced seizures (Skupien-Jaroszek et al, 2023). In rats, activity-driven Golgi disassembly is reversible and does not result in neuronal death, suggesting that it is a physiological response to fluctuations in network firing.

Molecular mechanisms driving activity-induced Golgi fragmentation and its consequences for neuronal function are unknown. Several groups of proteins were implicated in the control of the stacked Golgi structure in non-neuronal cells: cytoskeletal proteins, molecular tethers (GM130, p115), and Golgi matrix proteins (GRASP55, GRASP65). Depletion of these proteins results in Golgi fragmentation and has a strong impact on its secretory output (Wang et al, 2008; Xiang et al, 2013; Lavieu et al, 2014; Fourriere et al, 2016). For example, loss of GRASP55/65 led to an increase in cargo transport speed but impaired cargo glycosylation, suggesting that unstacking of the Golgi may serve as a tool to increase cell secretory output at the expense of the quality of cargo processing

(Wang et al, 2008; Xiang et al, 2013). In neurons, Golgi fragmentation upon prolonged increase in neuronal activity coincided with an increase in the number of Golgi satellites, suggesting that Golgi fragmentation may serve as a mechanism to increase processing of locally translated proteins (Govind et al, 2021).

Neuronal Golgi fragmentation is also a common cellular feature of many neurodegenerative diseases, such as Alzheimer's disease (AD) (Salehi et al, 1995; Stieber et al, 1996; Antón-Fernández et al, 2017), Parkinson's disease (Fujita et al, 2006), amyotropic lateral sclerosis (ALS) (Mourelatos et al, 1990; Gonatas et al, 1998), Creutzfeldt–Jakob disease (Sakurai et al, 2000) (Fig. 4). Characteristic changes in Golgi structure can be recapitulated in brains and spinal cords of mouse models of the corresponding diseases, indicating that this cellular phenotype is not an artifact caused by postmortem Golgi degradation (Liazoghli et al, 2005; Stieber et al, 2000; van Dis et al, 2014; Antón-Fernández et al, 2017). It is not clear if Golgi fragmentation is causative to neuronal death or rather is a consequence of other detrimental changes happening in neurodegenerative diseases, for example, aberrant protein aggregation.

Irreversible Golgi fragmentation also occurs in apoptosis (Aslan and Thomas, 2009); however, the neurodegeneration-associated loss of the Golgi structure does not appear to be secondary to apoptosis induction. Neurons with the fragmented Golgi lack other typical hallmarks of apoptosis, such as surface blebbing, shrinkage of the nucleus, or activation of caspase-3 (Zehr et al, 2004; Liazoghli et al, 2005; Gonatas et al, 2006; van Dis et al, 2014). Golgi fragmentation is also not a general reaction to neuronal injury, since it is not observed in response to neuronal deafferentation or to lesions of proximal axons of motor neurons (Mourelatos et al, 1994). A study in an ALS mouse model suggests that Golgi fragmentation precedes neuromuscular denervation and axon retraction, and happens in the early phase of disease progression (van Dis et al, 2014).

Proposed mechanisms of the Golgi fragmentation in neurodegeneration involve cytoskeletal rearrangements and vesicle trafficking defects (Haase and Rabouille, 2015). Pharmacological experiments support the role of the cytoskeleton and vesicle trafficking in Golgi integrity. Disruption of the microtubule network using microtubule

depolymerizing drugs or inhibition of the ER-to-Golgi transport using a fungal metabolite, brefeldin, induces Golgi fragmentation (Lippincott-Schwartz et al, 1989; Cole et al, 1996). Depolymerization of actin filaments also induces Golgi fragmentation, and this effect appears to be neuron-specific, as actin depolymerization in other cell types causes the opposite effect—Golgi compaction (Valderrama et al, 1998; Camera et al, 2003; Rosso et al, 2004; Lázaro-Diéguez et al, 2006). Exact triggering pathways leading to Golgi fragmentation likely depend on the disease type. For example, aberrant activity of dynactin/dynein, a microtubule motor complex disrupted in some ALS cases, causes dispersion of the Golgi (Burkhardt et al, 1997; Harada et al, 1998; Hoogenraad et al, 2001; Yadav and Linstedt, 2011). Microtubule-destabilizing protein stathmin-2 (STMN2), whose levels are decreased in the majority of ALS cases (Klim et al, 2019; Prudencio et al, 2020), was also implicated in the maintenance of Golgi structure (Strey et al, 2004; Bellouze et al, 2016). Overexpression of tau, the microtubule-associated protein aggregating in AD, induces Golgi fragmentation in mouse primary neurons, suggesting that tau accumulation causes changes in Golgi morphology (Liazoghli et al, 2005). Tau associates with Golgi membranes, and its accumulation may cause rearrangement of the Golgi-associated microtubules and consequently affect Golgi integrity (Farah et al, 2006). On the other hand, pharmacological or genetic induction of Golgi fragmentation itself causes an increase in tau hyperphosphorylation and subsequent aggregation (Jiang et al, 2014). Overexpression of the wild-type or a mutant form of amyloid precursor protein (APP), another protein aggregating in AD-affected brains, also causes Golgi fragmentation, possibly via the indirect effect of the APP-derived peptide Aβ on the phosphorylation of Golgi structural proteins such as GRASP65 (Joshi et al, 2014).

Interestingly, assembly of the Golgi into the characteristic ribbon coincides with the onset of embryonic development and cell differentiation in some ancestral animal species (Benvenuto et al, 2024). Conversely, undifferentiated mitotic cells have a disassembled Golgi complex. In AD, neurons undergo loss of terminal identity, dedifferentiate, and even attempt to re-enter the cell cycle (McShea et al, 1999). Thus, fragmented Golgi in AD neurons may be a hallmark of the attempted return to pluripotency and preparation for mitosis.

Consequences of Golgi fragmentation in neurodegeneration are just as elusive as its triggering mechanisms. In general, changes in the Golgi structure may initiate signaling pathways leading to apoptosis (Hicks and Machamer, 2005). In line with this, inhibition of Golgi fragmentation by overexpression of the Golgi matrix protein GRASP65 decreased death of cultured neurons exposed to NMDA (Nakagomi et al, 2008). In addition, Golgi fragmentation may affect the sorting and processing of the disease-relevant cargos, such as APP. It is unclear if these functional changes play a protective role or, conversely, contribute to the disease progression.

## Concluding remarks and open questions

In this review, we outlined some specialized features of the Golgi complex in neurons, such as polarization of the somatic Golgi, local Golgi elements in neurites, and the extensive molecular machinery for sorting of neuronal cargos. These features most likely evolved as a response to the particular architecture and functions of neuronal cells. As discussed above, polarization of the somatic Golgi contributes to the establishment and maintenance of the neuronal shape, local Golgi stations in neurites support secretory function of the remote synapses, and complex sorting machinery enables correct packaging and targeting of diverse neuronal cargos. Many questions concerning the structure and function of the specialized neuronal Golgi organelles are still open. For example, mechanisms of Golgi satellite formation, composition of these organelles, their potential heterogeneity, and functional contribution to the synaptic proteome are not known at the moment. Intricacies of the neuronal cargos sorting also await further investigation, especially considering the importance of this process for brain health. Arguably, one of the most intriguing questions is the structural and functional plasticity of the neuronal Golgi in response to changing network activity. How does acute and chronic elevation in neuronal activity affect the size and numbers of local Golgi elements? What are the molecular mechanisms of Golgi fragmentation resulting from the elevated synaptic activity? Do the same mechanisms contribute to the disassembly of neuronal Golgi in neurodegenerative diseases? What are the functional consequences of Golgi dispersion in health and pathological conditions? Considering fast progress in the field of cellular neuroscience, we expect some of these questions to be answered in the near future.

## Peer review information

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

## Acknowledgements

This work was supported by the Deutsche Forschungsgemeinschaft (DFG, German Research Foundation), the Excellence Strategy –EXC-2049-390688087 and the DFG FOR5228 RP4 to MM and DFG Rückkehrstipendium (SU 1131/2-1) to AS. We acknowledge support by the Open Access Publication Fund of Humboldt-Universität zu Berlin.

## Author contributions

**Aygul Subkhangulova**: Conceptualization; Funding acquisition; Visualization; Writing—original draft; Writing—review and editing. **Marina Mikhaylova**: Conceptualization; Funding acquisition; Visualization; Writing—original draft; Writing—review and editing.

## Funding

## Disclosure and competing interests statement

The authors declare no competing interests.

