## [Peer Review File · The EMBO Journal]

Golgi apparatus: adaptations to neuronal shape and functions

Aygul Subkhangulova and Marina Mikhaylova

Corresponding authors: Marina Mikhaylova (marina.mikhaylova@hu-berlin.de) , Aygul Subkhangulova (Aygul.Subkhangulova@dzne.de)

Review Timeline:

Submission Date:	25th Apr 25
Editorial Decision:	20th May 25
Revision Received:	21st Aug 25
Editorial Decision:	18th Sep 25
Revision Received:	8th Oct 25
Accepted:	14th Nov 25

Editor: Ieva Gailite

Transaction Report:

Dear Marina,

Thank you for submitting your review to The EMBO Journal. Your manuscript has now been seen by two reviewers, and I have attached their comments below.

As you will see, both reviewers appreciate the topic and the timeliness of the review. In addition, they provide several constructive points for the improvement, outlining aspects that need further clarification and a clearer discussion. They also find that the clarity of figures needs to be improved. Therefore, I invite you to submit a revised version of the manuscript with these suggestions incorporated.

From the editorial side, I have also included in the attachment further details on figure preparation for the final version.

Thank you for preparing such a timely review, and I look forward to receiving the revised version!

With best wishes,

Ieva

We realize that it is difficult to revise to a specific deadline. In the interest of protecting the conceptual advance provided by the work, we recommend a revision within 3 months (18th Aug 2025). Please discuss the revision progress ahead of this time with the editor if you require more time to complete the revisions.

Referee #1:

This review manuscript summarizes the current understanding of the architecture and function of the Golgi apparatus in neurons. The neuronal Golgi is uniquely adapted for the complex secretory pathways of this polarised cell and there have been recent advances (and surprises) that have enhanced the appreciation of the Golgi in neuronal cargo transport and synaptic function. The review is timely and nicely summarizes the role of different Golgi morphological states, the Golgi ribbon, Golgi outposts and Golgi satellites in neuronal signalling in healthy neurons and perturbations of the Golgi associated with neurological diseases. The topics are well researched and provide a different perspective from other recent reviews on this topic. I have a few suggestions that should be addressed to improve the quality of the review.

- The manuscript is generally well written although one major flaw is the omission of many definitive articles throughout the paper. I lost count of them! The paper needs to be thoroughly checked.
- The title of "Firing Golgi" is very obscure and should be removed or reworded

Major points:

1. The introduction (first page) could be abbreviated. The history of the identification of the Golgi has been covered in many articles and reviews and should be shortened which would allow the key focus of the review to be highlighted early.
2. Page 5. Regarding the statement that the somatic Golgi serves as an MTOC. The Golgi is known to be an MTOC in many cell types (eg papers by Kaverina) and this should be included to indicate it is not a unique feature of neurons.
3. I think Figure 2 is misleading. The species needs to be defined which is represented by this cartoon (rodents?). The distribution of Golgi outposts in dendrites in the figure suggests extensive dispersal through the dendritic tree, which is not correct in rodents. The outposts should predominantly be positioned in the proximal region of the dendrite and only very infrequently more distal. Also, the distribution of satellites within a single neuron I think is misleading. Overall, the figure

represents a summary of a population of neurons rather than a single neuron., which could readily give the wrong impression.

4. Golgi satellites, page 7. Golgi satellites remain very poorly defined. A more critical analysis of these structures would add further depth to the review. Golgi satellites are detected using one trans-Golgi network (TGN) marker, pGolt. It remains unclear whether other TGN markers eg. TGN golgins are associated with these structures. Also, to my knowledge, there is little (no) evidence that the glycosyltransferases of the Golgi stack are present in these satellite structures. There is evidence that Golgi satellites can mediate sialylation, however, it is conceptually difficult to see how ER-derived N-glycans can be processed into complex N-glycans exclusively from these structures. The final paragraph of the section on Golgi satellites "suggests GS may be solution to glycosylation of transmembrane proteins in distinct neurites" therefore seems very improbable. There has been a long debate on whether the TGN of mammalian cells can be considered independent from the Golgi stack and may be part of the endosomal system. This concept has been recently nicely summarised by Nakano (Frontiers in Cell Devel Biol 2022, 10). Of relevance is that the TGN is known to be dispersed from the Golgi stack proper following inflammation (Chen and Chen, Nature 2018; 564), indicating signalling events regulate the dispersal of this compartment and which could be relevant to the density of the GS following neuronal activation. It may be that this 'GS' compartment acts only in endocytic recycling. This suggestion was briefly mentioned in the manuscript but could be extended based on the ideas above. Also, the above comments can be integrated in the paragraph on mechanism of GS formation (page 8). The precise identity of these satellites clearly needs further work.

5. Page 13. Golgi fragmentation and the cytoskeleton. In addition to MT, actin also plays a major role in the Golgi architecture and can influence the fragmentation of the Golgi. The role of actin should be included

Minor points.

1. Other reviews on this topic include Kennedy and Hanus, Annu Rev Cell Dev Biol, 2019. 35: 543-566; Wang et al Front Mol Neurosci 2020, 13 and Valenzuela and Perez, (2015) Front. Neurosci. 9:358. doi: 10.3389/fnins.2015.00358. These should be included.
2. Golgi outposts- last line of second paragraph starting with "In Drosophila neurons" does not make sense as one compartment is not a stack.
3. End of 1st paragraph on Golgi Satellites. For iPSC-derived human neurons, it should be mentioned that analysis of pGolt structures was not been performed, therefore better to say that GS were not detected, however pGolt was not included in this study.
4. 2nd paragraph on Golgi Satellites. The description of the study by Hertrich et al 2024 (preprint bioRxiv) from the authors' lab needs more detail. Was the Golgi targeting region of the sialytransferase just the TM domain? If so, the interpretation may be complicated by the lipid composition of the membrane and the identity of the axonal structures may be unclear.
5. Page 10. Second last sentence in first paragraph beginning " On the contrary" is unclear
6. Page 11. First sentence of section on "Golgi sorting machinery and ...disorders" Not only mutations in sorting machinery but also the molecular tethers and Golgi matrix proteins (which are discussed in subsequent sections but should be included in this generic statement in the first sentence)

Referee #2:

Aygul Subkhangulova and Marina Mikhaylova have written a review focusing on the structure and function of the Golgi apparatus in neurons. I am very enthusiastic about this piece of work; it is timely, well-written, and stands out uniquely in a sub-field that is missing a comprehensive review. I am sure that this will end up being well-cited, and it was a pleasure to read.

I have some very minor comments:

- 1) the authors point out that drosophila do not have Golgi ribbons. This is true (as it is also true in some human cells like spinal ganglia)- and this is right, but there may be a broader evolutionary message here- I point the authors towards PMID:38428420, and they can adapt as they feel fit.
- 2) Cis/medial/trans should be in italics.
- 3) The authors state on page 4 that the neuron golgi produces "synaptic precursor vesicles"- i would like to see this referenced here as it was my understanding that synaptic vesicles were synthesised de novo at the synapse.
- 4) On page 5 the sentence "or even escape the route via the Golgi ('Golgi bypass') (Hanus et al, 2016; Bowen et al, 2017)" - was confusing to me for two reasons. A) escape what route? Golgi bypass is not via the Golgi but avoiding it? B) I think the evidence for the Golgi bypass is very weak. Perhaps it is not worth including in this review as it is not proven but gets propagated in contexts like this (but this is just my opinion- the authors can decide). The same is true for page 9, the Golgi bypass section- it is my feeling that this is really not established.
- 5) At the bottom of page 10 - "These fundamental sorting mechanisms are not unique to neurons and function in most other cell

types too.". I think this lacks some clarity. Yes, the fundamental mechanisms exist for parts in all cell types (TMD mediated, cytosolic tail). However, chromogranin-mediated condensation/aggregation is quite a novel concept that is not well understood and would only exist in cells with DCVs (i.e. exocrine, endocrine, neuroendocrine and neurons).

6) On page 11- I like this section on disease; however, I am not sure ARF1/ARFs is the best choice as it is also important for endosomal recruitment of coat protein complexes.

7) On page 2- "cargos are subjected to gradual processing"- this is not incorrect but perhaps a tad misleading as it is a sequential addition of glycans, and gradual processing may give the idea of a slow constant process.

8) I think it may be worth stating that there are many neuron subtypes, and they may be different. I know this is obvious to the writers, but this review will end up as a classic so getting ahead of a generalisation I think is wise here.

9) I think the Golgi in figure 4 came from bioicons - the authors should be aware of the required attribution- "Attribution - Give credit to Servier in the figure caption or acknowledgements, link the license and indicate changes. Do not suggest endorsement."

10) The authors mention AP-3 but fail to mention that there is a neuronal-specific isoform of the beta subunit of the complex (AP-3B), which may be relevant in that section. Nice review here: PMID:17567262

11) Some papers the authors may have missed that they can read and include if they wish- PMIDs: 26500481, 28238793, 28485842, 32317144, 1906810, 18197413 - and the chapter "Biogenesis of secretory granules" By JD Castle ISBN 9780750337717

Point-by-point reply to referees

We thank the reviewers for taking time to assess our work and helping us to improve the manuscript. We truly appreciate their constructive feedback. Below is our point-by-point response to reviewers' questions and suggestions with the original reviewers' text in brown and our response in black.

Referee #1:

This review manuscript summaries the current understanding of the architecture and function of the Golgi apparatus in neurons. The neuronal Golgi is uniquely adapted for the complex secretory pathways of this polarised cell and there have been recent advances (and surprises) that have enhanced the appreciation of the Golgi in neuronal cargo transport and synaptic function. The review is timely and nicely summaries the role of different Golgi morphological states, the Golgi ribbon, Golgi outposts and Golgi satellites in neuronal signalling in healthy neurons and perturbations of the Golgi associated with neurological diseases. The topics are well researched and provide a different perspective from other recent reviews on this topic. I have a few suggestions that should be addressed to improve the quality of the review.

- The manuscript is generally well written although one major flaw is the omission of many definitive articles throughout the paper. I lost count of them! The paper needs to be thoroughly checked.

Reply: Thank you for pointing this out. We thoroughly checked the manuscript and added a number of important citations to the revised version, among them PMID 38428420, 32553155, 29631154, 35285533, 21502307, 35573670, 33098763, as well as all the reviews suggested by the Reviewer 1 in Minor Point 1 and Reviewer 2 in Point 11.

- The title of "Firing Golgi" is very obscure and should be removed or reworded

Reply: We propose a new title for the revised manuscript "Golgi apparatus: adaptations to neuronal shape and functions".

Major points:

1. The introduction (first page) could be abbreviated. The history of the identification of the Golgi has been covered in many articles and reviews and should be shortened which would allow the key focus of the review to be highlighted early.

Reply: We agree. We shortened the part on the history of Golgi identification and research and refer the readers to the reviews on this topic (PMID: 9695800, 9610881, 21079007; lines 7-9).

2. Page 5. Regarding the statement that the somatic Golgi serves as an MTOC. The Golgi is known to be an MTOC in many cell types (eg papers by Kaverina)) and this should be included to indicate it is not a unique feature of neurons.

Reply: Thank you for pointing it out, we included this information in the chapter on somatic Golgi (lines 137-139).

3. I think Figure 2 is misleading. The species needs to be defined which is represented by this cartoon (rodents?). The distribution of Golgi outposts in dendrites in the figure suggests extensive dispersal through the dendritic tree, which is not correct in rodents. The outposts should predominantly be positioned in the proximal region of the dendrite and only very infrequently more distal. Also, the distribution of satellites within a single neuron I think is misleading. Overall, the figure represents a summary of a population of neurons rather than a single neuron., which could readily give the wrong impression.

Reply: The reviewer is right, the figure intends to depict specifically a rodent pyramidal neuron, and we now modified the depiction of Golgi outposts and satellites accordingly. The Golgi outposts are now positioned in the proximal region of the primary dendrite and at the dendrite branching point, as suggested by studies in primary hippocampal neurons (PMID 16337914, 25802147). The distribution of satellites is shown as suggested by the immuno-EM and fluorescent stainings of brain sections from pGolt-expressing mice (PMID 37355986). GS are shown as vesicles/ vesicle clusters uniformly distributed along dendrites of CA1 pyramidal neurons. Localization of satellites is not restricted to or more prominent in proximal dendrites. Satellites are also present throughout the axon, where they are frequently localized to pre-synaptic boutons (doi: <https://doi.org/10.1101/2024.05.08.593268>).

4. Golgi satellites, page 7. Golgi satellites remain very poorly defined. A more critical analysis of these structures would add further depth to the review. Golgi satellites are detected using one trans-Golgi network (TGN) marker, pGolt. It remains unclear whether other TGN markers eg. TGN golgins are associated with these structures. Also, to my knowledge, there is little (no) evidence that the glycosyltransferases of the Golgi stack are present in these satellite structures. There is evidence that Golgi satellites can mediate sialylation, however, it is conceptually difficult to see how ER-derived N-glycans can be processed into complex N-glycans exclusively from these structures. The final paragraph of the section on Golgi satellites "suggests GS may be solution to glycosylation of transmembrane proteins in distinct neurites" therefore seems very improbable. There has been a long debate on whether the TGN of mammalian cells can be considered independent from the Golgi stack and may be part of the endosomal system. This concept has been recently nicely summarised by Nakano (Frontiers in Cell Devel Biol 2022, 10). Of relevance is that the TGN is known to be dispersed from the Golgi stack proper following inflammation (Chen and Chen, Nature 2018; 564), indicating signalling events regulate the dispersal of this compartment and which could be relevant to the density of the GS following neuronal activation. It may be that this 'GS' compartment acts only in endocytic recycling. This suggestion was briefly mentioned in the manuscript but could be extended based on the ideas above. Also, the above comments can be integrated in the paragraph on mechanism of GS formation (page 8). The precise identity of these satellites clearly needs further work.

Reply: We expanded the chapter on GS to address the reviewer's concerns and incorporate their suggestions (pages 7-9). We agree there are still a lot of open questions and controversy on the

identity and functions of GS, which we point out in the revised manuscript. We expanded the paragraph on the potential role of GS in endocytic recycling and included the hypothesis on the endosomal/ TGN identity of GS and the view on the TGN as the organelle independent of the main Golgi body and rather belonging to the endosomal system (*lines 283-297*). In this respect, it is interesting to note that most GS do not colocalize with the TGN-resident golgin 97 and TGN marker TGN38 (*lines 239-240*). There is also little colocalization of GS with the typical early endosomal markers (*lines 294-295*). As for the potential role of GS in cargo glycosylation, we replaced the sentence '*GS may be solution to glycosylation of transmembrane proteins in distant neurites*' with '*GS may carry out selective glycosylation reactions of transmembrane cargos synthesized or recycled in distal neurites. GS may also regulate local cargo sorting...*' (*lines 312-313*). The potential role of GS in cargo glycosylation is supported by the current experimental evidence:

- several late-acting Golgi glycosylation enzymes (e. g. MAN2A1, GALNT2, ST8SIA2, ST3GAL5) were localized to GS by light microscopy (*lines 257-258*).
- GS are enriched in maturely glycosylated proteins as shown by labeling of GS with specific lectins and azido monosaccharides (*lines 261-268*)
- manipulation of GS numbers coincided with changes in the glycosylation status of neuronal proteome (*lines 270-279*)

We fully agree with the reviewer that more studies are needed to clarify the functional role of GS, as we also point out in the manuscript.

5. Page 13. Golgi fragmentation and the cytoskeleton. In addition to MT, actin also plays a major role in the Golgi architecture and can influence the fragmentation of the Golgi. The role of actin should be included

Reply: We agree. We included information on the role of actin in Golgi integrity (*lines 497-500*).

Minor points.

1. Other reviews on this topic include Kennedy and Hanus, *Annu Rev Cell Dev Biol*, 2019. 35: 543-566; Wang et al *Front Mol Neurosci* 2020, 13 and Valenzuela and Perez, (2015) *Front. Neurosci.* 9:358. doi: 10.3389/fnins.2015.00358. These should be included.

Reply: We included these citations in the manuscript (*lines 92-93*).

2. Golgi outposts- last line of second paragraph starting with "In *Drosophila* neurons" does not make sense as one compartment is not a stack.

Reply: We rephrased this sentence as following (*lines 196-197*): *In Drosophila neurons, cis-, medial- and trans-compartments of GO are often disconnected and form complete stacks only at dendritic branch points (Zhou et al, 2014).*

3. End of 1st paragraph on Golgi Satellites. For iPSC-derived human neurons, it should be mentioned that analysis of pGolt structures was not been performed, therefore better to say that GS were not detected, however pGolt was not included in this study.

Reply: This is correct, we added this information to the first paragraph (*lines 243-245*).

4. 2nd paragraph on Golgi Satellites. The description of the study by Hertrich et al 2024 (preprint bioRxiv) from the authors' lab needs more detail. Was the Golgi targeting region of the sialyltransferase just the TM domain? If so, the interpretation may be complicated by the lipid composition of the membrane and the identity of the axonal structures may be unclear.

Reply: In fact, the reporter used in this study was the full-length sialyltransferase St3Gal5 fused to the fluorescent tag (GFP), and not just the Golgi-targeting region of St3Gal5 as we stated in the original manuscript. We apologize for this error and thank the reviewer for bringing this issue to our attention. This reporter has already been used for visualization of GS in the original study by Mikhaylova *et al.*, where it was shown to colocalize with pGolt (Figure 2C, Mikhaylova *et al.* 2016). We corrected the information on the reporter in the revised manuscript (*lines 249-251*)

5. Page 10. Second last sentence in first paragraph beginning " On the contrary" is unclear

Reply: We rephrased this sentence as following (*lines 345-346*): SV cargos are not detected on late endosomes by EM (Tao-Cheng, 2020).

6. Page 11. First sentence of section on "Golgi sorting machinery and ...disorders" Not only mutations in sorting machinery but also the molecular tethers and Golgi matrix proteins (which are discussed in subsequent sections but should be included in this generic statement in the first sentence)

Reply: Thank you for pointing this out. We completely agree and corrected the first sentence and the title of this chapter (*line 398*).

Referee #2:

Aygul Subkhangulova and Marina Mikhaylova have written a review focusing on the structure and function of the Golgi apparatus in neurons. I am very enthusiastic about this piece of work; it is timely, well-written, and stands out uniquely in a sub-field that is missing a comprehensive review. I am sure that this will end up being well-cited, and it was a pleasure to read.

Reply: Thank you for your encouraging feedback!

I have some very minor comments:

1) the authors point out that drosophila do not have Golgi ribbons. This is true (as it is also true in some human cells like spinal ganglia)- and this is right, but there may be a broader evolutionary message here- I point the authors towards PMID:38428420, and they can adapt as they feel fit.

Reply: Thank you for referring us to this article. We find it particularly interesting in light of the potential function of the ribbon structure in cell differentiation, as suggested by this study. We now included the mentioning of the ribbon evolution in the introduction (*page 2, lines 17-20*). We also added a paragraph in the chapter on the Golgi fragmentation (*page 14, lines 516-521*), where we speculate about the link between ribbon disassembly and neuronal dedifferentiation in neurodegenerative diseases.

2) *Cis/medial/trans* should be in italics.

Reply: Corrected.

3) The authors state on page 4 that the neuron golgi produces "synaptic precursor vesicles" - i would like to see this referenced here as it was my understanding that synaptic vesicles were synthesised de novo at the synapse.

Reply: By "synaptic precursor vesicles" we mean transport carriers up to 300 nm in size, which carry presynaptic proteins from the somatic Golgi to presynapses. These vesicles are formed at the Golgi (in contrast to synaptic vesicles, which, as the reviewer correctly pointed out, are formed directly at the presynaptic sites). To avoid any confusion, we added the definition and citations to the main text (*lines 104-105*). We also changed from 'synaptic precursor vesicles' to 'synaptic vesicle precursors'. Both terms are commonly used in the field (PMID 33098763, 32553155, 31601744).

4) On page 5 the sentence "or even escape the route via the Golgi ('Golgi bypass') (Hanus et al, 2016; Bowen et al, 2017)" - was confusing to me for two reasons. A) escape what route? Golgi bypass is not via the Golgi but avoiding it? B) I think the evidence for the Golgi bypass is very weak. Perhaps it is not worth including in this review as it is not proven but gets propagated in contexts like this (but this is just my opinion- the authors can decide). The same is true for page 9, the Golgi bypass section- it is my feeling that this is really not established.

Reply: We agree with the reviewer that the Golgi bypass in neurons is far away from being well-characterized. However, this trafficking route has been described and (arguably) better characterized in other cell types (e.g. for CFTR, cystic fibrosis transmembrane conductance regulator, and some cilia proteins in epithelial cells - PMID 11799116, 21884936, 21321097). Therefore, we feel it is worth mentioning the possibility of the Golgi bypass in neurons, especially since there is some evidence for it in the literature. We rephrased the sentence on page 5 and describe Golgi bypass as "the delivery of transmembrane proteins from the ER to the plasma membrane without those proteins passing through the Golgi" (*lines 164-166*). On the same page, we refer the reader to some reviews, where this topic is discussed in detail, in neurons and non-neuronal cells (PMID 21441587, 23377655, 29631154). We removed the separate section on the Golgi bypass from our manuscript since we feel that the cited reviews cover this topic sufficiently.

5) At the bottom of page 10 - "These fundamental sorting mechanisms are not unique to neurons and function in most other cell types too.". I think this lacks some clarity. Yes, the fundamental

mechanisms exist for parts in all cell types (TMD mediated, cytosolic tail). However, chromogranin-mediated condensation/aggregation is quite a novel concept that is not well understood and would only exist in cells with DCVs (i.e. exocrine, endocrine, neuroendocrine and neurons).

Reply: We agree and rephrased this sentence as following: "Some of the mechanisms listed above, such as AP-mediated sorting, are not unique to neurons and function in most other cell types; while others, i.e. chromogranin-mediated DCV formation are restricted to secretory cells (neurons, exocrine, endocrine and neuroendocrine cells)" (lines 384-387).

6) On page 11- I like this section on disease; however, I am not sure ARF1/ARFs is the best choice as it is also important for endosomal recruitment of coat protein complexes.

Reply: We agree. That is also true for adaptor protein complexes or RAB GTPases. We now added a new paragraph in this chapter, where we mention that many discussed proteins function not only at the Golgi but also on other endomembranes (page 12, lines 443-446). We also added examples of other disease-associated proteins that function specifically at the Golgi (such as GM130, COG complex, VPS13 proteins).

7) On page 2- "cargos are subjected to gradual processing"- this is not incorrect but perhaps a tad misleading as it is a sequential addition of glycans, and gradual processing may give the idea of a slow constant process.

Reply: We agree and replaced the word "gradual" with "sequential".

8) I think it may be worth stating that there are many neuron subtypes, and they may be different. I know this is obvious to the writers, but this review will end up as a classic so getting ahead of a generalisation I think is wise here.

Reply: Point well taken. We added a short paragraph on the variety of neuronal types (page 4, lines 84-90).

9) I think the Golgi in figure 4 came from bioicons - the authors should be aware of the required attribution- "Attribution - Give credit to Servier in the figure caption or acknowledgements, link the license and indicate changes. Do not suggest endorsement."

Reply: Correct, we apologize for missing the attribution. We added it to the figure legend.

10) The authors mention AP-3 but fail to mention that there is a neuronal-specific isoform of the beta subunit of the complex (AP-3B), which may be relevant in that section. Nice review here: PMID:17567262

Reply: Thank you for pointing this out. We added this information to the chapter "Sorting of neuronal cargos at the somatic Golgi" (page 11, lines 391-394).

11) Some papers the authors may have missed that they can read and include if they wish- PMIDs: 26500481, 28238793, 28485842, 32317144, 1906810, 18197413 - and the chapter "Biogenesis of secretory granules" By JD Castle ISBN 9780750337717

Reply: We thank the reviewer for suggesting these papers. We incorporated most of them in the revised paragraph on DCV formation (page 10, lines 351-361).

Dear Marina,

Thank you for submitting the revised version of your review to The EMBO Journal. Your manuscript has now been seen by both original reviewers, who find that most of their comments have been addressed successfully. Reviewer #1 still has some minor points that I would like to ask you to incorporate in the final version.

I will now forward your figures to our collaborating graphic editor Luk Cox for final stylistic adaptation.

Please also add a "Disclosure and competing interests statement" before References. Further information about this section can be found here: <https://www.embopress.org/page/journal/14602075/authorguide#conflictsofinterest>.

Please let me know if you have any questions, and I look forward to working with you to polish up the final version of your manuscript!

With best wishes,

Ieva

We realize that it is difficult to revise to a specific deadline. In the interest of protecting the conceptual advance provided by the work, we recommend a revision within 3 months (17th Dec 2025). Please discuss the revision progress ahead of this time with the editor if you require more time to complete the revisions.

Referee #1:

The revised manuscript by Aygul Subkhangulova and Marina Mikhaylova has addressed my concerns and suggestions on the scientific issues and, together with the suggestions from reviewer 2, have improved this very timely review.

There was one issue in my earlier feedback which was misinterpreted by the authors. I had previously indicated that "The manuscript is generally well written although one major flaw is the omission of many definitive articles throughout the paper. I lost count of them! The paper needs to be thoroughly checked." The authors misinterpreted "definitive articles" as citations, however, I was referring to omission of "definitive articles" of the English grammar, eg. "the". Maybe I didn't make this clear. Examples as follows with the definitive article now included: third last sentence of the Abstract "and discuss the principles" First sentence of Introduction: "The Golgi apparatus.." Line 18: "The ribbon-like Golgi.." Line 44 ", the Golgi performs". Line 53 : "The Golgi size..." There are many others. I will leave this issue for the authors to discuss with the Editor.

1. The new text (in red) needs to be proof-read to improve the writing in some instances. For example line, 269, should read "Helix pomatia agglutinin, a lectin which binds to O-glycans". Line 408; ... "often most prominently affect" change to "affect": and line 409 change "coined with the term "golgipathies" to "coined golgipathies".
2. There is inconsistency in the use of lowercase/capital for the term "golgins". Generally, the field uses lowercase.
3. On re-reading I note that in the section on "Sorting of neuronal cargos at the somatic Golgi" that AP-1 and AP-3 are discussed (line 381). However, AP-4 has been overlooked which is an important adaptor at the TGN and the loss of AP-4 in neurons results in neurological disorders. I don't think it is necessary to include a discussion on AP4, rather perhaps give AP-1 and AP-3

as examples of adaptor protein complexes in neurons, to remove the mis-conception that AP-1 and AP-3 maybe the only relevant adaptor complexes at the TGN.

4. The Golgi apparatus has been abbreviated to GA in a few instances (lines 125, 128, 147 (pages 4, 5). Please correct.

5. All the cis, medial trans terms need to be in italics (only some are).

Referee #2:

I was positive about this in my first review and my minor comments have all be addressed. I support the publication of this and look forward to sharing and citing it.

Point-by-point reply to Referees

We thank the referees for their valuable feedback on the revised manuscript. As suggested by the Reviewer 1, we have now addressed some writing- and English grammar-related issues that evaded our attention in the first revision round. Below is the list of the introduced changes with the original reviewers' text in blue and our response in black.

Referee #1:

The revised manuscript by Aygul Subkhangulova and Marina Mikhaylova has addressed my concerns and suggestions on the scientific issues and, together with the suggestions from reviewer 2, have improved this very timely review.

There was one issue in my earlier feedback which was misinterpreted by the authors. I had previously indicated that "The manuscript is generally well written although one major flaw is the omission of many definitive articles throughout the paper. I lost count of them! The paper needs to be thoroughly checked." The authors misinterpreted "definitive articles" as citations, however, I was referring to omission of "definitive articles" of the English grammar, eg. "the". Maybe I didn't make this clear. Examples as follows with the definitive article now included; third last sentence of the Abstract "and discuss the principles" First sentence of Introduction:"The Golgi apparatus.." Line 18: "The ribbon-like Golgi.." Line 44 ", the Golgi performs". Line 53 : "The Golgi size..." There are many others. I will leave this issue for the authors to discuss with the Editor.

We apologize for misinterpreting the reviewer's comment. We have corrected the usage of definite articles throughout the manuscript text.

1. The new text (in red) needs to be proof-read to improve the writing in some instances. For example line, 269, should read "Helix pomatia agglutinin, a lectin which binds to O-glycans". Line 408; ..."often most prominently affect" change to "affect": and line 409 change "coined with the term "golgiopathies" to "coined golgiopathies".

We proofread the manuscript text and corrected it for errors and inconsistencies in sentence structure, word choice, grammar and punctuation. We followed the suggestions of the reviewer on the above-listed phrases.

2. There is inconsistency in the use of lowercase/capital for the term "golgins". Generally, the field uses lowercase.

Corrected.

3. On re-reading I note that in the section on "Sorting of neuronal cargos at the somatic Golgi" that AP-1 and AP-3 are discussed (line 381). However, AP-4 has been overlooked which is an important adaptor at the TGN and the loss of AP-4 in neurons results in neurological disorders. I

don't think it is necessary to include a discussion on AP4, rather perhaps give AP-1 and AP-3 as examples of adaptor protein complexes in neurons, to remove the misconception that AP-1 and AP-3 maybe the only relevant adaptor complexes at the TGN.

Thank you for pointing it out. We have not mentioned AP-4 in lines 392-398 because this complex, in contrast to AP-1 and -3, does not have any neuron-specific subunits. In the context of that paragraph, AP-1 and -3 served as examples of complexes with specialized neuronal composition:

..some sorting proteins are either enriched or expressed exclusively in neurons. For example, adaptor protein complexes AP-1 and AP-3, components of transport vesicle coat, though being expressed ubiquitously, have a distinct composition in neurons in comparison to other cell types...

That said, we totally agree that AP-4 is an important coat protein with respect to its localization to the TGN and association with neurological disease. Therefore, we now included mentioning of AP-4 in the chapter on golgipathies (lines 438-440).

4. The Golgi apparatus has been abbreviated to GA in a few instances (lines 125, 128, 147 (pages 4, 5). Please correct.

Corrected.

5. All the cis, medial trans terms need to be in italics (only some are).

Corrected.

Referee #2:

I was positive about this in my first review and my minor comments have all be addressed. I support the publication of this and look forward to sharing and citing it.

Thank you for positive evaluation of our review.

Dear Marina,

I sincerely apologise for the slow processing of your revised review article due to the high number of primary article submissions that we have to treat with priority. I am now pleased to inform you that your review has been accepted for publication in the EMBO Journal.

Before we forward your manuscript to our typesetters, I would like to propose some minor edits for style and clarity throughout the manuscript. Please take a look in the attached file and accept or modify as needed.

I would also like to propose the following blurb that will accompany the title of your article in our online table of contents:

"This review summarises our current knowledge of the unique properties of the Golgi in the neurons and its contribution to neuronal development, function and disease."

Your manuscript will be processed for publication by EMBO Press. It will be copy edited and you will receive page proofs prior to publication.

You will soon be contacted by Springer Nature to sign your publishing license. When you login to the customer service website, please use the following token to waive the article publication charges:

[removed]

Should you experience any difficulty, please email publishing@embo.org.

If you have any questions, please do not hesitate to contact me or the Editorial Office. Thank you once more for this thought-provoking contribution to The EMBO Journal, and I look forward to receiving your input on the final textual edits.

With best wishes,

leva

leva Gailite, PhD
Senior Scientific Editor
The EMBO Journal
Meyerhofstrasse 1
D-69117 Heidelberg
Tel: +4962218891309
i.gailite@embojournal.org

Please note that it is The EMBO Journal policy for the transcript of the editorial process (containing referee reports and your response letters) to be published as an online supplement to each paper. If you should prefer removal of any referee-only figures included in the point-by-point response(s), e.g. because they may still be used for future publication or because they have been reproduced from published work by others, please do let us know immediately via response email.

More information is available here: https://www.embopress.org/transparent-process#Review_Process